# University of Kentucky measurements of wind, temperature, pressure and humidity in support of LAPSE-RATE using multi-site fixed-wing and rotorcraft UAS

Sean C. C. Bailey[1], Michael P. Sama[2], Caleb A. Canter[1,*], L. Felipe Pampolini[2], Zachary S. Lippay[1], Travis J. Schuyler[3,**], Jonathan D. Hamilton[1,***], Sean B. MacPhee[1,****], Isaac S. Rowe[1], Christopher D. Sanders[2], Virginia G. Smith[1,*****], Christina N. Vezzi[1], Harrison M. Wight[1,***], Jesse B. Hoagg[1], Marcelo I. Guzman[3], and Suzanne Weaver Smith[1]

[1]Department of Mechanical Engineering, University of Kentucky, Lexington, Kentucky 40506, USA
[2]Biosystems and Agricultural Engineering, University of Kentucky, Lexington, Kentucky 40546, USA
[3]Department of Chemistry, University of Kentucky, Lexington, Kentucky 40506, USA
[*]Present Address: AHS, Hodgenville, Kentucky 42748, USA
[**]Present Address: Atmospheric Turbulence and Diffusion Division, Air Resources Laboratory, National Oceanic and Atmospheric Administration, Oak Ridge, Tennessee 37830, USA
[***]Present Address: Department of Aerospace Engineering Sciences, University of Colorado Boulder, Boulder, Colorado 80303, USA
[****]Present Address: Bastian Software Solutions, Louisville, Kentucky 40223, USA
[*****]Present Address: Department of Aerospace and Ocean Engineering, Virginia Polytechnic Institute and State University, Blacksburg, Virginia 24061, USA

**Correspondence:** Sean Bailey (sean.bailey@uky.edu)

**Abstract.** In July 2018, unmanned aerial systems (UAS) were deployed to measure the properties of lower atmosphere within the San Luis Valley, an elevated valley in Colorado, USA as part of the Lower Atmospheric Profiling Studies at Elevation – a Remotely-piloted Aircraft Team Experiment (LAPSE-RATE). Measurement objectives included detailing boundary-layer transition, canyon cold-air drainage, and convection initiation within the valley. Details of the contribution to LAPSE-RATE made by University of Kentucky are provided here, which include measurements by seven different fixed-wing and rotorcraft UAS totaling over 178 flights with validated data. The data from these coordinated UAS flights consist of thermodynamic and kinematic variables (air temperature, humidity, pressure, wind speed and direction) and include vertical profiles up to 900 m above the ground level and horizontal transects up to 1500 m in length. These measurements have been quality controlled and are openly available in the Zenodo LAPSE-RATE community data repository (https://zenodo.org/communities/lapse-rate/), with the University of Kentucky data available at (https://doi.org/10.5281/zenodo.3701845 (Bailey et al., 2020).)

## 1 Introduction

This manuscript discusses the systems and contribution of the University of Kentucky researchers to the Lower Atmospheric Profiling Studies at Elevation – a Remotely-piloted Aircraft Team Experiment (LAPSE-RATE) campaign (de Boer et al., 2020b) conducted from July 13, 2018 through July 19, 2018 in the San Luis Valley in Colorado, USA. In this campaign

teams of unmanned aerial system (UAS) based atmospheric science research teams cooperated with researchers from the National Center for Atmospheric Research (NCAR), the National Severe Storms Laboratory (NSSL) and the National Oceanic and Atmospheric Administration (NOAA) (de Boer et al., 2020a) to conduct UAS sensor intercomparison, along with separate days of UAS and ground observations focused on boundary-layer transition, canyon cold-air drainage, and convection initiation. This paper describes the contribution from University of Kentucky researchers towards the campaign objectives, including a description of the systems used and discussion of the data acquired during the LAPSE-RATE campaign (de Boer et al., 2020b).

University of Kentucky's research in unmanned aircraft systems (UAS) flight testing evolved over more than 450 flights conducted to evaluate performance of deployable-wing unmanned aircraft (Jacob et al., 2005, 2007; Thamann et al., 2015), before expanding into atmospheric turbulence (Witte et al., 2016, 2017), formation and flight control (Mullen et al., 2016; Heintz et al., 2019; Wellman and Hoagg, 2018; Heintz and Hoagg, 2019; Lippay and Hoagg, 2019), measurement of atmospheric gas concentrations (Schuyler and Guzman, 2017; Schuyler et al., 2019a, b) and as participants in the CLOUDMAP program (Jacob et al., 2018). The CLOUDMAP program focused on the development of UAS technologies for meteorology and atmospheric science and resulted in advancement of capabilities for atmospheric observations with UAS, as demonstrated via multi-university flight campaigns (Smith et al., 2017), validation experiments (Barbieri et al., 2019), and through observations of the surface layer transitions during the 2017 total eclipse (Bailey et al., 2019).

For the LAPSE-RATE campaign, the University of Kentucky deployed four fixed-wing UAS and three-rotorcraft UAS, which measured pressure, temperature, relative humidity, horizontal wind magnitude, vertical wind magnitude, and direction. The UAS were used to measure these thermodynamic and kinematic variables at altitudes up to 900 m above ground level (AGL). A surface flux tower was also deployed, capable of measuring net solar radiation, soil temperature, ground heat flux, surface wind and turbulence, and the near-surface temperature gradient. These assets were deployed to seven different sites over the course of LAPSE-RATE, measuring at up to five sites simultaneously to contribute to scientific objectives targeting convection initiation, boundary layer transition and cold-air drainage. Over the course of the week, the team conducted over 178 flights yielding vetted observation data. Details about the systems used, their deployment, and quality-control checks applied to the data are described in the remainder of the manuscript, with the data files openly available (Bailey et al., 2020).

The following sections of this paper include descriptions of the diverse UAS and ground systems used by the University of Kentucky during LAPSE-RATE, followed by details of measurement locations and completed flights. Data processing and quality control specifics are included, which provide information needed by those wanting to use this data. Examples of some of the features observed within the data are highlighted as well.

## 2 Description of the UAS and ground systems

### 2.1 BLUECAT5 UAS

Four fixed-wing UAS were used by the University of Kentucky during the LAPSE-RATE measurement campaign. These UAS, of which one is shown in Figure 1(a), were constructed from Skywalker X8 airframes, modified to introduce semi-autonomous operation by integrating a 3DR Pixhawk autopilot, and ruggedized by strengthening wing spars, skinning the

aircraft, adding Kevlar landing skids, and shielding to minimize the ingestion of dirt into the motor. Referred to as the Boundary Layer Unmanned Experiment for the Characterization of Atmospheric Turbulence generation five (BLUECAT5) UAS (Witte et al., 2017), the aircraft had endurance of up to 45 minutes with 20 m/s cruise speeds and were catapult launched and skid landed. The four aircraft used here are identified as BCT5B, BCT5C, BCT5D and BCT5E.

In the configuration used for LAPSE-RATE, the instrumentation allowed for the measurement of three-components of wind, as well as pressure, temperature and humidity. Airspeed information used by the autopilot was acquired using a 30 cm long Pitot-static tube extending from the nose of the aircraft. In addition, the Pitot-static tube was used to provide a reference static pressure for a custom-manufactured five-hole pressure probe used to measure the velocity vector of the air relative to the aircraft. Pressure from each port of the the five-hole probe were acquired using TE Connectivity, Switzerland 4515-DS5A002DP differential pressure transducers with a 0.5 kPa range. Analog output from the sensors was digitized at 16-bit resolution at a rate of 400 Hz with a 200 Hz passive RC low-pass filter used for anti-aliasing. Digitization was performed using a MCC-DAQ USB-1608FS-PLUS multi-function data acquisition system controlled by a Kangaroo portable PC .

Each five-hole probe was calibrated for directional response using a 0.6 m × 0.6 m wind tunnel. The calibration followed a standard calibration technique outlined by Treaster and Yocum (1978) which was implemented following the results presented by Wildmann et al. (2014). During these studies, the probe tubing length was optimized to provide a frequency response on the order of 100 Hz. At the typical cruise speed of BLUECAT5, this frequency response translates to a spatial measurement resolution of approximately 0.2 m. Additional wind and water tunnel studies verified that positioning the measurement volume of the probe 18 cm upstream of the nose of the aircraft was sufficient to minimize interference effects from the airframe (Witte et al., 2017).

Six-degree of freedom position and rate information was provided by a VN-300 manufactured by VectorNav. The VN-300 provided a heading accuracy of ±0.3° and pitch/roll accuracy of ±0.1° with ground velocity accuracy of ±0.05 m/s. The orientation information was sampled at 200 Hz by custom software running on the Kangaroo PC.

Pressure, temperature and humidity were measured using an International Met Systems iMet-XQ UAV sensor. The iMet pressure sensor provides a ±1.5 hPa accuracy for pressure, with humidity sensor supports a full 0 - 100 %RH range at ± 5 %RH accuracy with a resolution of 0.7 %RH. The iMet-XQ temperature sensor provides a ± 0.3 °C accuracy with a resolution of 0.01 °C up to a maximum of 50 °C. The stated response times of these sensors are on the order of 10 ms for pressure, 5 s for humidity and 2 s for temperature in still air, with the iMet-XQ UAV system sampling these sensors at 1 Hz. The iMet-XQ sensor was mounted on top of the aircraft with the thermistor exposed to the airflow but shielded by a 3D printed ABS arch designed to protect it from heating via solar radiation.

To determine the wind-velocity vector, the velocity of the air measured by the five-hole probe in an aircraft-fixed frame of reference is transformed into an Earth-fixed frame of reference using inertial velocity provided by the VN-300. Following this transformation, the aircraft's velocity relative to the ground subtracted, leaving the three-component wind vector (Axford, 1968; Lenschow, 1972; Broxmeyer and Leondes, 1964). Further details are available in Witte et al. (2017) with additional corrections applied in to correct for probe orientation bias. During this post-processing step, all data were re-sampled to 200 Hz and aligned

in time via cross-correlation of the signals measured by the different systems. However, the data from these aircraft should only be considered to be resolved to 100 Hz for wind, 1 Hz for pressure, 0.5 Hz for temperature and 0.2 Hz for humidity.

## 2.2 SOLOW UAS

Pressure, temperature, humidity and wind measurements were also conducted using a 3DRobotics SOLO quadrotor identified here as the SOLOW and shown in Figure 1(b). This UAS was capable of semi-autonomous flight through a Pixhawk autopilot and had approximately 10 minutes of endurance for each battery, but near-continuous operations were possible by keeping the sufficient batteries charged and available to ensure that the aircraft could be returned to flight with approximately 3 to 5 minutes on the ground for a battery change. Used for vertical profiling measurements, this aircraft was typically operated with 2 m/s

descent and 3 m/s ascent velocities, which optimizes the number of vertical profiles which could be obtained from a single battery charge while maintaining the thermodynamic data within a nominally 2.5 m measurement resolution.

For measuring the pressure, temperature and relative humidity, the UAS was equipped with an International Met Systems iMet-XQ-2 UAV sensor, in a custom mount below one of the rotors to ensure sufficient aspiration of the sensors with solar radiation shielding of the humidity and thermistor provided by the mount. Wind speed and direction sensing was provided by a

95 TriSonica-Mini sonic anemometer manufactured by Applied Technologies with manufacturer-provided accuracy of $\pm 0.1$ m/s and $\pm 1°$ in wind speed and direction respectively. The anemometer was mounted on a 0.38 m carbon fiber post above the main body of the rotorcraft with the optimal mast height determined through laboratory tests consisting of increasing the post length until the UAS motors running at full speed did not result in a change in anemometer reading. Additional laboratory calibration was conducted in a 0.6 m $\times$ 0.6 m wind tunnel to account for blockage effects from the sensor housing with calibration applied

*a posteriori*. Although nominally a three-component anemometer, it had an acceptance cone for vertical winds of $\pm 30°$ to the vertical which, when exceeded, contaminated the horizontal components of velocity as discussed in Section 4.

Digital output from the sonic anemometer was logged at 10 Hz during flight using a Slerj RS232 data logger and analyzed *a posteriori* following the same procedures utilized for the fixed-wing aircraft, but with the aircraft position and orientation information extracted at 10 Hz from the autopilot flight logs. Corresponding pressure, temperature and humidity data were

105 interpolated to 10 Hz, but should only be considered resolved to 1 Hz for pressure, 0.5 Hz for temperature and 0.2 Hz for humidity.

## 2.3 S1000 UAS

Pressure, temperature, humidity and wind measurements were also conducted using a DJI S1000 octocopter, identified here as the S1000, shown in Figure 1(c). The S1000 was modified for semi-autonomous operation by implementing a Pixhawk

autopilot, and was capable of approximately 20 minutes of flight time for each battery pack. Used for vertical profiling measurements, this aircraft was typically operated with 1 m/s descent and ascent velocities, which optimizes the altitude which could be obtained for flight measuring a single vertical profile on a single battery charge while minimizing the spatial resolution of the thermodynamic data to nominally 1 m.

Pressure temperature and relative humidity measurements were provided by an International Met Systems iMet-XQ system. These systems log data at 1 Hz, with a stated response and accuracy of 10 ms, $\pm 1.5$ hPa for pressure; 1 s, $\pm 0.3$ °C for temperature; and 0.6 s, $4 \pm 0.5$ %RH for relative humidity. The sensors were located under one of the aircraft's rotors for sensor aspiration in a housing designed to minimize the impact of solar radiation on the sensing.

To measure wind speed and direction an R.M. Young, USA Model 81000 ultrasonic anemometer was mounted on a mast with its measurement volume located 0.55 m above the rotor plane, at the centerline of the aircraft. The Young 81000 could measure wind speeds up to 40 m/s at a resolution of 0.01 m/s with accuracy of $\pm$ 0.05 m/s. The anemometer was set to output velocity and sonic temperature data as analog signals, which were digitized at a nominal rate of 70 Hz by a Mayhew labs 16-bit analog-to-digital converter controlled by an Arduino embedded computer.

As with the SOLOW UAS, the data were processed *a posteriori* to subtract aircraft motion from the wind data. To do so the aircraft position and orientation information, sonic anemometer data, and IMET-XQ data were interpolated to 10 Hz. Although pressure, temperature and humidity data were interpolated to 10 Hz, it should only be considered resolved to 1 Hz for this aircraft.

## 2.4   M600P UAS

An additional UAS was used for pressure, temperature and humidity profiling, based on the DJI M600P hexacopter platform. This aircraft, shown in Figure 1(d), used the manufacturer provided batteries and autopilot, allowing for semi-autonomous operation with approximately 20 minutes of flight time. Here we identify this UAS as the M600P.

Instrumentation on this aircraft was provided by two International Met iMet-XQ2 sensing systems. These sensors are the same type as described above in the description of the S1000; however, on the M600P these two were mounted below opposite rotors to maintain weight and balance. Data from both sensors were logged independently at 1 Hz.

Unlike with the other UAS, the autopilot data were not available for use in post-processing the IMET-XQ2 sensor data. Instead, the intrinsic IMET-XQ2 GPS and altimeter data were used to identify UAS position and altitude information, with the pressure, temperature and humidity data from one sensor interpolated to sample times corresponding to the other sensor to unify the data stream. The intrinsic IMET-XQ2 GPS was a UBlox CAM-M8 with vertical accuracy of 12 m and response time of 1 s.

## 2.5   Flux Tower

Additional ground-based measurements were provided by a 2 m surface flux tower shown in Figure 2. The tower was equipped with a three-component sonic anemometer (Campbell Scientific CSAT-3) for measuring wind speed and direction. Sensor accuracy was between $\pm 2\%$ and $\pm 6\%$ with $\pm 0.08$ m/s bias precision. Output from the anemometer was logged at 20 Hz via RS232 using a Kangaroo PC portable computer mounted in a weatherproof enclosure.

The tower was equipped with a Campbell Scientific E+E Electronik EE181 digital temperature and humidity sensor at 2 m having accuracy of $\pm 0.2$ °C and $\pm 2.3$ %RH. At 1.5 m and 0.75 m the tower had two additional Campbell Scientific CS215

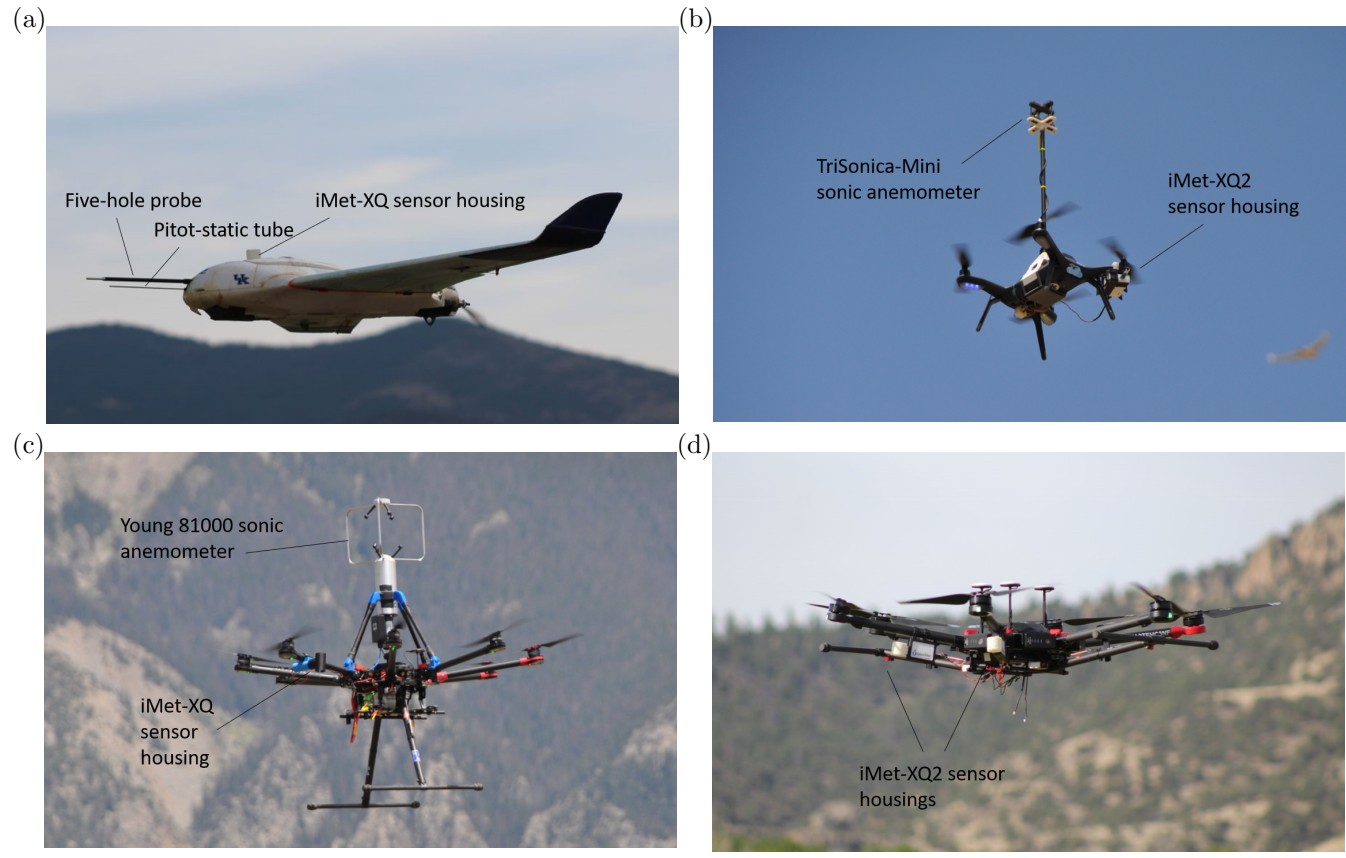

**Figure 1.** Photographs of (a) BCT5, (b) SOLOW, (c) S1000, and (d) M600P showing location of sensors on each aircraft.

digital temperature and humidity sensors ($\pm 0.4$ °C, $\pm 4$ %RH). All temperature and humidity sensors were housed in a solar radiation shield and logged every 3 s via a Campbell Scientific CR1000X measurement and control data logger.

Additional sensors on the flux tower logged every 3 s included a Setra 278 digital barometer, Kipp and Zonen NR-LITE2 net radiometer, two Hukeseflux HFP01 soil heat flux plates, a Campbell Scientific CS655 water content reflectometer, and Campbell Scientific TCAV averaging soil thermocouple probe. All sensors were factory calibrated within one year of use, although intercomparison measurements in a laboratory environment revealed that the EE181 sensor had a consistent $0.5$ °C bias which was removed from the measurements reported here.

## 3 Description of measurement location(s), flight strategies and completed flights/sampling

The systems described above were deployed at different locations throughout the LAPSE-RATE measurement campaign. These locations are identified as Leach Airfield, Echo, Foxtrot, Kilo, Oscar, Saguache Airfield, and Poison Gulch. Approximate latitude and longitude of these locations are provided in Table 1 with flight locations graphically illustrated in Figure 3. Flight

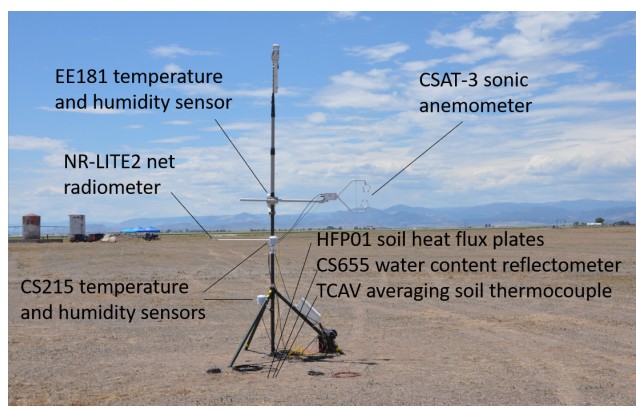

**Figure 2.** Photograph flux tower in position at Leach Airport illustrating sensor configuration.

**Table 1.** Latitude and longitude of operation locations in World Geodetic System 84 (WGS 84) decimal degrees.

| Location | Latitude | Longitude |
|---|---|---|
| Leach Airfield | 37.779291 | -106.039235 |
| Echo | 37.705015 | -106.093502 |
| Foxtrot | 37.646877 | -105.926981 |
| Kilo | 37.712359 | -105.947122 |
| Oscar | 37.86782 | -105.92853 |
| Saguache Airfield | 38.09404 | -106.16269 |
| Poison Gulch | 38.157308 | -106.280973 |

locations and UAS dispositions varied by day of operations and measurement objective with UAS measurements being conducted in the morning and early afternoon. Aircraft dispositions and measurement times are outlined below and summarized in Table 2. The flux tower was located at Leach Airfield and was measuring continuously until the afternoon of July 18, 2018.

The Leach Airfield, Echo, Foxtrot, Kilo and Oscar sites were centrally located in the San Luis Valley, consisting largely of agricultural land having minimal elevation change and loose, dry soil. The Poison Gulch and Saguache Airfield sites were located at two points along a narrow valley leading into the larger San Luis Valley. The Poison Gulch site being a location where the valley was narrow with steep valley walls, whereas the Saguache Airfield site was located closer to the mouth of the valley, where the valley was much broader and having shallower elevation changes.

Civil twilight during the measurement period started between 5:24 MDT on July 15 and 5:27 MDT on July 19, with the corresponding sunrise times being between 5:54 MDT and 5:57 MDT. Sunsets for the same period were between 20:26 MDT and 20:24 MDT with the civil twilight ending between 20:57 MDT and 20:54 MDT.

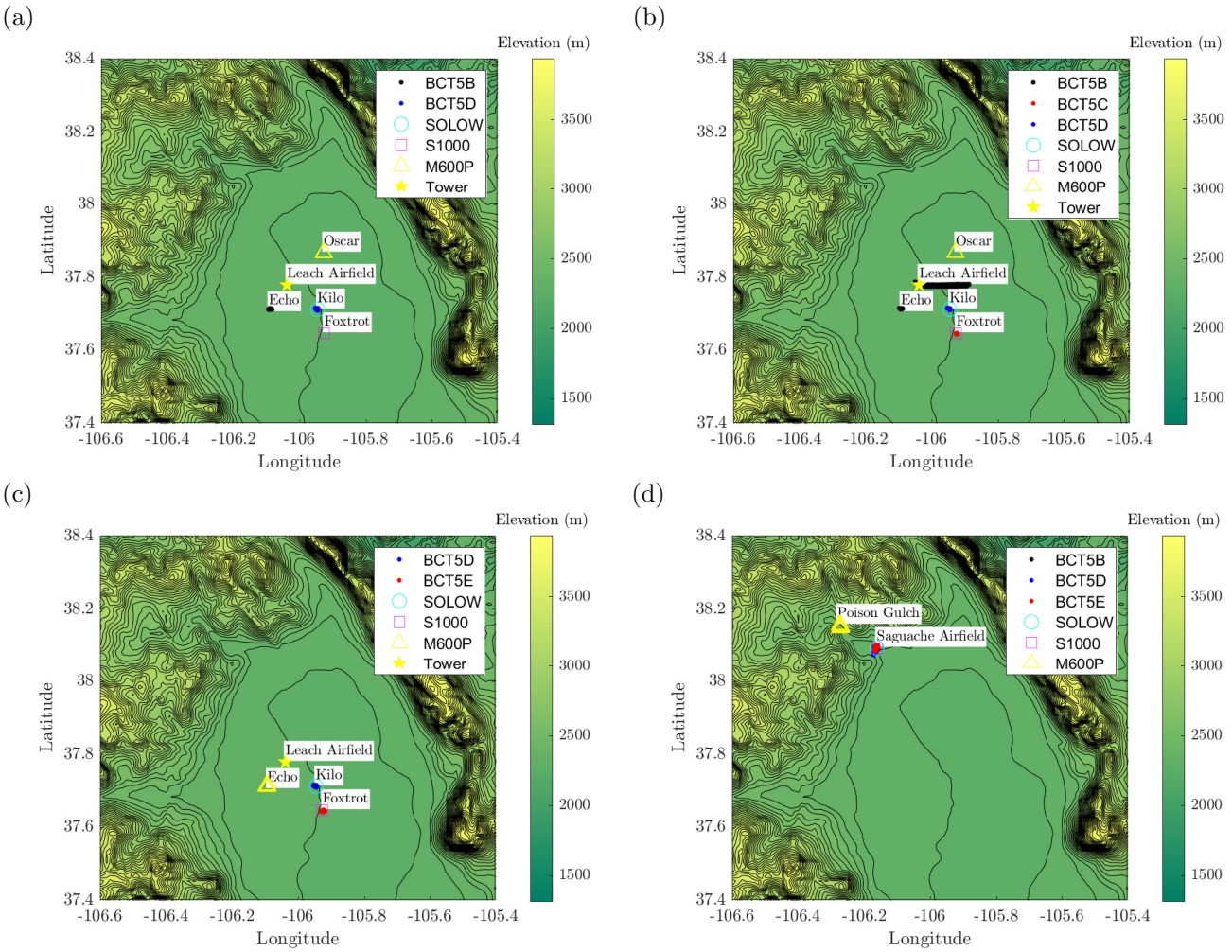

**Figure 3.** Topographic map of Northern half of San Luis Valley, Colorado, USA showing location of tower and aircraft flights conducted by BCT5, S1000, SOLOW, and M600P aircraft on: (a) July 15, 2018; (b) July 16, 2018; (c) July 18, 2018; and (d) July 19, 2018. Contour lines indicate 50 m change in elevation.

**Table 2.** Measurement systems location. (P) indicates measurement of a vertical profile, (T) indicates a horizontal transect measurement.

| Day | Leach Airfield | Echo | Foxtrot | Kilo | Oscar | Saguache Airfield | Poison Gulch |
|---|---|---|---|---|---|---|---|
| July 15, 2018 (7:30-14:30 MDT) | Tower | BCT5B (P) | S1000 (P) | BCT5D (P) SOLOW (P) | M600P (P) | | |
| July 16, 2018 (8:00-14:30 MDT) | Tower BCT5B (T) | BCT5B (P) | BCT5C (P) S1000 (P) | BCT5D (P) SOLOW (P) | M600P (P) | | |
| July 18, 2018 (7:00-12:00 MDT) | Tower | M600P (P) | BCT5E (P) S1000 (P) | BCT5D (P) SOLOW (P) | | | |
| July 19, 2018 (5:30-11:00 MDT) | | | | | | BC5TB (T) BC5TD (T) BC5TE (T) SOLOW (P) S1000 (P) | M600P (T) |

All flights were conducted legally under either FAA Part 107 regulations, which limited flight operations to 121 m or below, or under the University of Kentucky's Federal Aviation Authority (FAA) Certificate of Authority (COA) 2018-WSA-1730-COA effective from July 13, 2018 to July 22, 2018. The COA authorized operation of small UAS weighing less than 55 pounds (25 kg) and operating at speeds of less than 87 knots (45 m/s) in Class E and G airspace below 914 m AGL and not exceeding 3657 m above mean sea level (MSL) in the vicinity of Alamosa County, CO under the jurisdiction of Denver Air Route Traffic Control Center (ARTCC). Standard COA provisions were applied including those for airworthiness, operations, flight crew, safety, NOTAMs, reporting, and registration. Special provisions were also necessary for coordination and deconfliction of operations of multiple organizations. Rather than authorizing different overlapping operating areas defined by the pre-planned positioning for each organization, discussions among the organizations, FAA and Denver ATC led to definition of one common area encompassing the entire LAPSE-RATE plan of operations. With this, additional coordination and deconfliction became necessary with airports and several intersecting Military Training Routes (MTRs). Two NOTAM subareas were defined so that one or both could be issued with the required 24-hr notice depending on the selected weather question that would be pursued the next day. Emergency contingency procedures for lost links, communications and other potential anomalies also had special provisions due to the proximity of operations of the various organizations.

### 3.1 July 15, 2018

July 15, 2018 was scheduled for measurements investigating convection initiation. To support this objective, vertical profiling flights measuring kinematic and thermodynamic parameters were conducted at Echo, Foxtrot, Kilo and Oscar with BC5TB at

Echo, the S1000 at Foxtrot, BCT5D and the SOLOW at Kilo and the M600P at Oscar. The nominal flight profile for these flights was for the fixed wing UAS to conduct 5 m/s, 200 m diameter spiraling ascents and descents to 900 m AGL with the rotorcraft ascending and descending vertically to 300 m AGL. The S1000 measured a single profile per flight ascending/descending at 1 m/s whereas the SOLOW completed two ascents/descents at 3 m/s upward and 2 m/s downward velocity. The UAS repeated these patterns from takeoff until their battery life was expended, at which point they would land and have their batteries changed. The nominal tempo for these patterns was to conduct these flights every once per hour. At Kilo, where two aircraft were available, the flights were staggered such that one aircraft was launched every half hour. A figure showing the height, $z$, of each aircraft as a function in time is presented in Figure 4(a). This figure serves as a graphical representation of the time each aircraft flew over the course of the day.

On this day, the wind speed and direction was not measured by the SOLOW aircraft due to data acquisition issue which was not diagnosed until after flights had concluded. In addition, the S1000 conducted limited profiles at Foxtrot due to overheating of the flight batteries.

### 3.2 July 16, 2018

July 16, 2018 was also scheduled for measurements investigating of convection initiation and flight operations were expected to be a repeat of July 15, 2018 with vertical profiling flights measuring of kinematic and thermodynamic parameters. UAS were located at Echo, Foxtrot, Kilo and Oscar with BC5TB at Echo, BCT5C and the S1000 at Foxtrot, BCT5D and the SOLOW at Kilo and the M600P at Oscar. However, it was discovered that the Notice to Airman (NOTAM) was not active for any operation locations (excepting Oscar), so flight operations could not be conducted under the COA regulations, requiring flights to be restricted to 120 m and below. As a result, it was decided that the fixed-wing operations could be better utilized and their flight crews were re-tasked to conduct horizontal east-west transects with BCT5B to attempt to capture any fronts descending from the Sangre de Cristo Mountains to the east. Two 13 km long transects were conducted along the path initiating at Leach airport (see Figure 3(b)) in conjunction with a ground vehicle from University of Nebraska-Lincoln.

To compensate for BC5TD being re-tasked, the cadence of the SOLOW operations were increased to nominally one 1 m/s ascent/descent profile measurement to 120 m every 15 minutes. However, due to continued battery issues, the S1000 was not able to maintain this cadence and separation between 1 m/s ascent/descent profile measurements to 120 m was between 30 and 60 minutes. A graphical representation of the time each aircraft flew over the course of the day is presented in Figure 4(b). Note that some discrepancies exist in the M600P profiles exist, caused by poor GPS performance from the onboard systems on this day.

### 3.3 July 18, 2018

July 18, 2018 was scheduled for measurements investigating boundary layer transition. As a result, profiling flights were again conducted following the same flight patterns as on July 15. However, due to concerns with potential low-altitude manned aircraft flights through the VR-413 airway, the M600P was re-tasked to Echo. A fixed-wing and rotorcraft UAS were located

at Foxtrot and Kilo, with BCT5E and the S1000 at Foxtrot, and BCT5D and the SOLOW at Kilo. Flight operations on this day largely went as scheduled, with the cadence presented in Figure 4(c).

## 3.4  July 19, 2018

July 19, 2018 was scheduled for measurements of the cold air drainage into the valley. For this study, all aircraft were operating in the narrow valley leading into Saguache with the flight locations and patterns indicated in Figure 5(a) for Poison Gulch and Figure 5(b) for Saguache Airfield. Most aircraft were operating at Saguache Airfield, with the M600P operating further up the mouth of the valley, in Poison Gulch.

At Poison Gulch, the M600P was flying 600 m long horizontal transects at 25 m AGL and every 50 m up to 250 m AGL
at 10 m/s every 30 minutes to capture the flow entering the mouth of the valley. To measure the flow approaching the mouth of the valley, the remainder of the aircraft were positioned at Saguache Airfield. To capture the horizontal distribution of the thermodynamic and kinematic variables, the fixed-wing aircraft measured nominally 1700 m long transects at four different altitudes with the rotorcraft flying vertical profiles. The SOLOW rotorcraft was conducting near-continuous profiles at 2 to 3 m/s up to 100 m from 5:30 MDT to 11:00 MDT with the S1000 profiling at 1 m/s every hour from 6:00 MDT to 11:00 MDT
up to 300 m. The transects were measured at a nominal altitude of 100 m AGL, 150 m AGL and 200 m and 400 m AGL. Due to a longer preparation time for the fixed-wing aircraft, full operations did not initiate until 7:00 MDT, with flights continuing to 10:30 MDT. The actual flight times and altitudes for all aircraft are presented in Figure 4(d).

## 4  Data Processing and Quality Control

Inspection of the raw data from these flights revealed several instances were the instrumentation was not functioning properly.
As noted above, the SOLOW sonic anemometer was not powered during the July 15 measurements, and no wind data were recovered from those flights. On July 17, the pressure, temperature and humidity from the SOLOW only seems to have recorded first two flights. In addition, a power issue was also discovered with the S1000 anemometer whereby the anemometer would shut down approximately halfway through the profile. This problem persisted throughout the campaign. Finally, several fixed-wing flights encountered overheating and malfunction of the embedded computer, resulting in a loss of wind data for these
flights. Where these faults were observed, the values in the files have been replaced with values of -9999.9.

Furthermore, erroneous readings were observed for the SOLOW anemometer during profiling flights which were caused by $\pm 30°$ vertical acceptance angle violations. Wind data from this system should only be considered to be reliable when the aircraft was descending due to a lower vertical velocity during descent. These instances have been replaced in the data files with -9999.9. In addition, slight misalignment between ground and relative air velocity measurements introduces large biases
in the wind estimates provided by the BCT5 aircraft during periods of significant acceleration, specifically during takeoff, landing and sharp turns. These biases were removed using the approach described in Al-Ghussain and Bailey (2020). However, wind data from the time periods when these aircraft are entering or leaving their measurement profile should be considered unreliable due to accelerations which exceeded the ability of the approach to detect and remove.

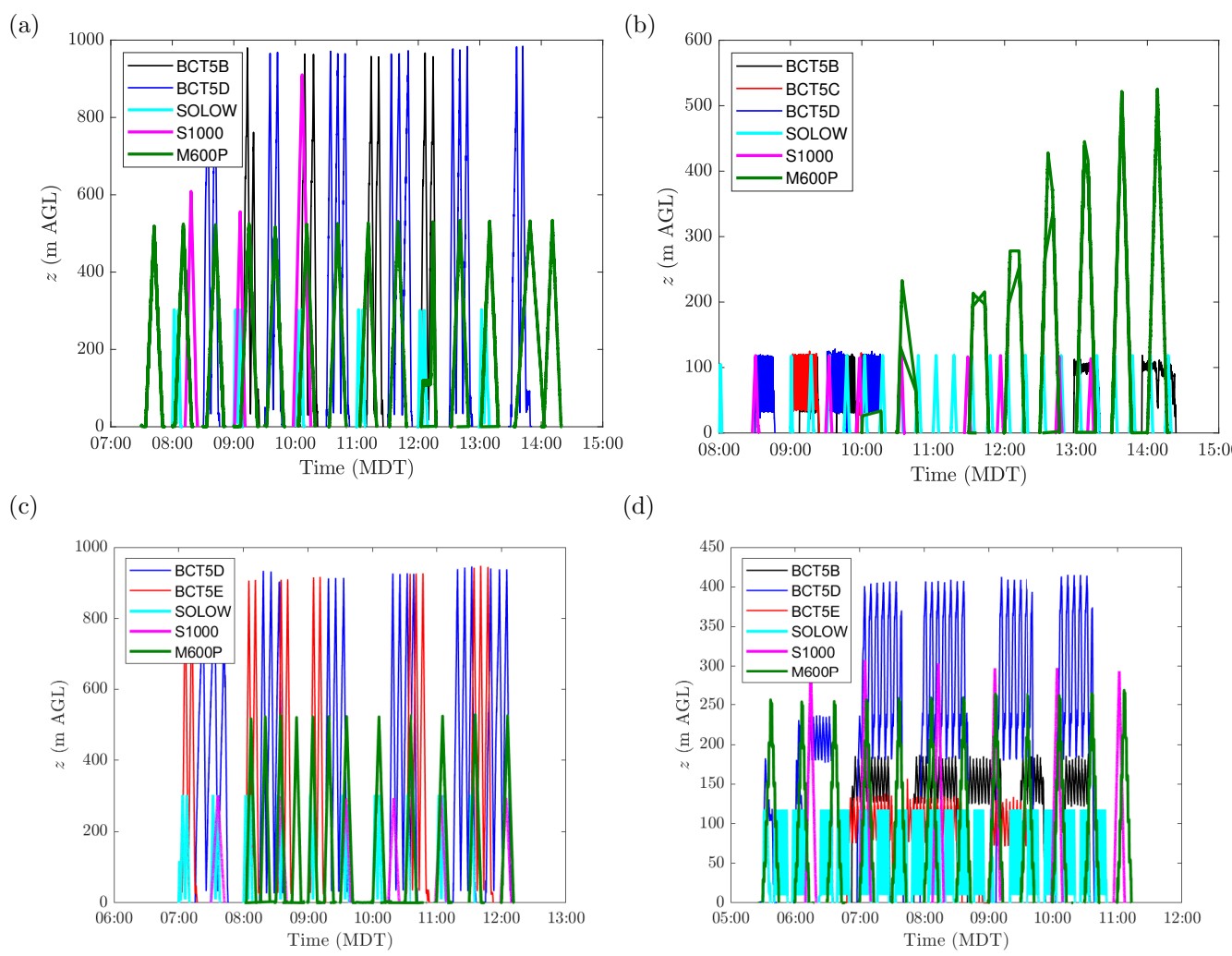

**Figure 4.** Altitude above ground level of each aircraft as a function of time for: (a) July 15, 2018; (b) July 16, 2018; (c) July 18, 2018; and (d) July 19, 2018.

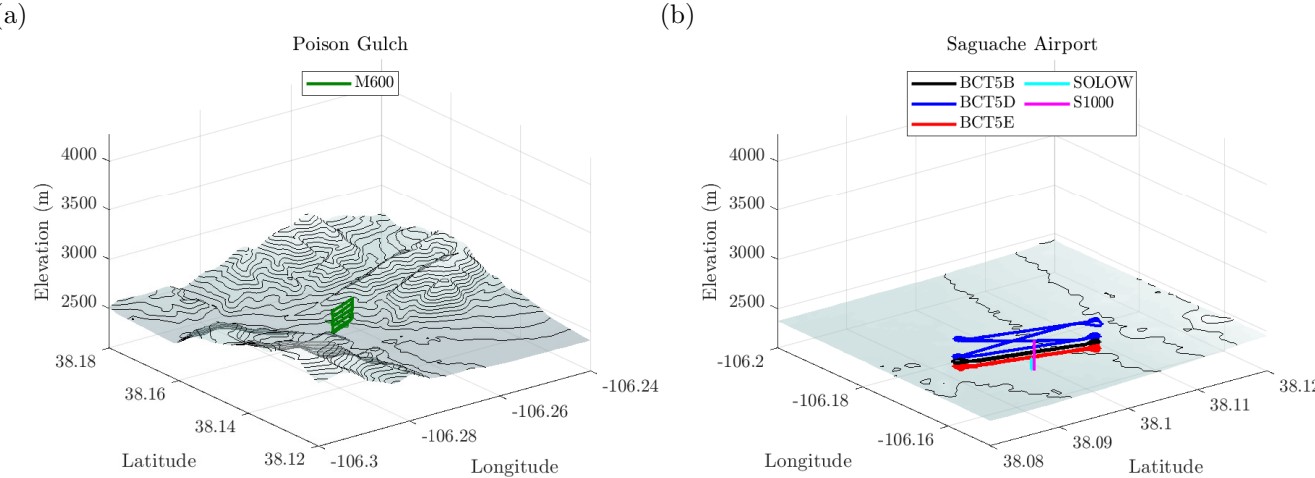

**Figure 5.** Flight profiles for flights conducted by all aircraft on July 19, 2018. Flight trajectories at Poison Gulch shown in (a), with (b) showing the flight trajectories at Saguache Airfield.

As noted in the individual system descriptions, several laboratory calibrations were conducted prior to the flight campaign, most notably for the systems for measuring wind. In addition, as part of an intercomparison study, a series of flights were conducted on July 14, 2018 near a ground-based reference tower. The results of this study, published in Barbieri et al. (2019), indicated an overestimation of the winds measured by the BCT5 aircraft by up to 10 % to 20%. Additional laboratory calibrations of the S1000 anemometer indicated that the horizontal winds were underestimated by 25%.

These biases were found to be constant throughout the campaign and independent of weather conditions and flight profile, as intrinsically validated by intercomparison between all systems and the Automated Surface Observing System (ASOS) using the co-located flights at Saguache Airfield. As the wind direction was found to be unaffected, they have therefore been uniformly removed from the data files by applying them equally to both horizontal components of velocity. However the magnitude of these corrections do reflect the level of uncertainty in the respective wind measurements. In addition, utilizing the Barbieri et al. (2019) intercomparison study findings, we also find that the temperature has an absolute confidence level of $\pm 1\ ^\circ$C, the relative humidity values have an absolute confidence level of approximately $\pm 5\%$ and the pressure values have an absolute confidence level of $\pm 2$ hPa. An overview of the systems, their instruments and corresponding resolution and estimated uncertainty is provided in Table 3.

## 5 Overview of data

This section provides a general overview of the data provided by the systems described in Section 2 during the LAPSE-RATE campaign from the mornings of July 15, 16, 18 and 19. An overview of the prevailing synoptic and mesoscale conditions during the week is provided in de Boer et al. (2020b), as well as de Boer et al. (2020a).

**Table 3.** Overview of measurement systems and associated uncertainties. Measured quantities are: $P$ pressure; $T$ temperature; $RH$ relative humidity; $u, v$ and $w$ wind velocity components in horizontal and vertical direction respectively; Net IR net solar radiation; and $VWC$ volumetric water content.

| System | Measurement | Instrument | Resolution | Est. Uncertainty |
|---|---|---|---|---|
| BCT5 | $P$ | iMet-XQ | 0.02 hPa | $\pm 2$ hPa |
|  | $T$ | iMet-XQ | 0.01 $^\circ$C | $\pm 1\,^\circ$C |
|  | $RH$ | iMet-XQ | 0.7 %$RH$ | $\pm 5$ %$RH$ |
|  | $u, v, w$ | Custom 5HP | Analog | $\pm 20\%$ |
| SOLOW | $P$ | iMet-XQ2 | 0.01 hPa | $\pm 2$ hPa |
|  | $T$ | iMet-XQ2 | 0.01 $^\circ$C | $\pm 1\,^\circ$C |
|  | $RH$ | iMet-XQ2 | 0.1 %$RH$ | $\pm 5$ %$RH$ |
|  | $u, v$ | Trisonica Mini | 0.1 m/s | $\pm 25\%$ |
| S1000 | $P$ | iMet-XQ | 0.02 hPa | $\pm 2$ hPa |
|  | $T$ | iMet-XQ | 0.01 $^\circ$C | $\pm 1\,^\circ$C |
|  | $RH$ | iMet-XQ | 0.7 %$RH$ | $\pm 5$ %$RH$ |
|  | $u, v$ | Young 81000 | 0.01 m/s | $\pm 25\%$ |
| M600P | $P$ | iMet-XQ2 ($\times 2$) | 0.01 hPa | $\pm 2$ hPa |
|  | $T$ | iMet-XQ2 ($\times 2$) | 0.01 $^\circ$C | $\pm 1\,^\circ$C |
|  | $RH$ | iMet-XQ2 ($\times 2$) | 0.1 %$RH$ | $\pm 5$ %$RH$ |
| Tower | $P$ | Setra 278 | 0.01 hPa | $\pm 0.5$ hPa |
|  | $T$ | EE181 (2 m) | 0.01 $^\circ$C | $\pm 0.5\,^\circ$C |
|  | $T$ | CS215 (0.75 m & 1.5 m) | 0.01 $^\circ$C | $\pm 0.5\,^\circ$C |
|  | $RH$ | EE181 (2 m) | 0.01 %$RH$ | $\pm 2.3$ %$RH$ |
|  | $RH$ | CS215 (0.75 m & 1.5 m) | %$RH$ | $\pm 4$ %$RH$C |
|  | $u, v, w$ | CSAT-3 | 0.08 m/s | $\pm 6\%$ |
|  | Net IR | NR-Lite2 | 0.1 W/m$^2$ | 30 W/m$^2$ |
|  | Soil Heat Flux | HFP01 ($\times 2$) | 0.1 W/m$^2$ | $\pm 15\%$ |
|  | Soil Ave. $T$ | TCAV | 0.01 $^\circ$C | $\pm 1\,^\circ$C |
|  | $VWC$ | CS655 | 0.001 (m$^3$/m$^3$) | $\pm 3\%$ |
|  | Soil $T$ | CS655 | 0.01 $^\circ$C | 0.5 $^\circ$C |

The measurements made by the 2 m flux tower are presented in Figure 6. Figure 6(a) shows that surface winds from July 15 to July 18 were variable in direction, generally below 5 m/s but picking up in the late afternoon with the development of convective boundary layer conditions, occasionally reaching velocity magnitudes exceeding 10 m/s. Convective thunderstorm activity occurred in the afternoons as well, resulting in additional wind gusts corresponding to their outflows. A consistent diurnal cycle was observed for July 15 through July 18 in heat flux (Figure 6(b)), temperature, $T$ (Figure 6(c)), relative humidity, $RH$ (Figure 6(d)) and volumetric water content $VWC$ (Figure 6(e)). The calm winds in the mornings corresponded to elevated humidity, with the $100\ \%RH$ observed on the morning of July 18 corresponding to early morning fog formation. Near-surface air temperatures ranged from 10 °C to 30 °C with soil temperatures following the same cycle but elevated to between 20 °C and 50 °C. The volumetric water content was low, reflecting the dry soil conditions prevalent in the San Luis Valley (evident in Figure 2).

Mean vertical profiles from all temperature measurements made are presented in Figure 7 with the corresponding velocity magnitude profiles presented in Figure 8. These profiles were produced by bin-averaging the results at 10 m intervals for each flight. The temperature profiles from July 15, which were measured between 7:30 MDT and 14:30 MDT and are presented in in Figure 7(a), show typical transition behavior, with the lower-temperature, early-morning profiles showing non-monotonic altitude dependence up to 800 m AGL, whereas the later afternoon profiles consist of the characteristic lapse-rate of a well-mixed convective boundary layer. Over the same time period, the wind profiles shown in Figure 8(a) indicate the presence of calm winds of generally less than 3 m/s magnitude. Wind profiles were essentially the same on July 16 between 8:00 MDT and 14:30 MDT, shown in Figure 8(b), although the altitude restrictions described in Section 3.2 limited measurements of winds to the lowest 100 m. Likewise, the temperature profiles shown in Figure 7(b) show a consistent lapse-rate throughout the morning. More interesting boundary layer transition behavior was evident on July 18 between 7:00 MDT and 12:00 MDT. The temperature profiles shown in Figure 7(c) indicate the presence of multiple inversions during the measurements made earlier in the morning, which evolve into a well-mixed lapse-rate as surface temperatures increase. The corresponding wind profiles shown in Figure 8(c) show that, although relatively calm winds were evident near the surface in during the entire measurement period, significantly stronger winds and wind shear existed above 500 m AGL. Finally, the measurements made in the Saguache valley on July 19 between 5:30 MDT and 11:00 MDT show in Figure 7(d) the expected reversal of gradient in the temperature profiles associated with the earlier morning boundary layer behavior as it transitions to mixed-layer conditions over the course of the morning. As expected over the complex terrain of the valley, the winds were variable as shown in Figure 8(d), with mean velocity peak near 6 m/s caused by a transient density-driven flow event.

## 6 Example profiles

Of the 178 measurements made, as noted in the previous section some of the most interesting profiles were measured on the morning of July 18, 2018. On this morning the weather was fair, having less than 3/8 cloud cover, with surface winds varying from calm to 2.5 m/s. Surface temperature increased from 10 °C to 25 °C with a dew point dropping from 5 °C to -1 °C throughout the morning. However the boundary layer was stratified into multiple layers evident in each of the measured

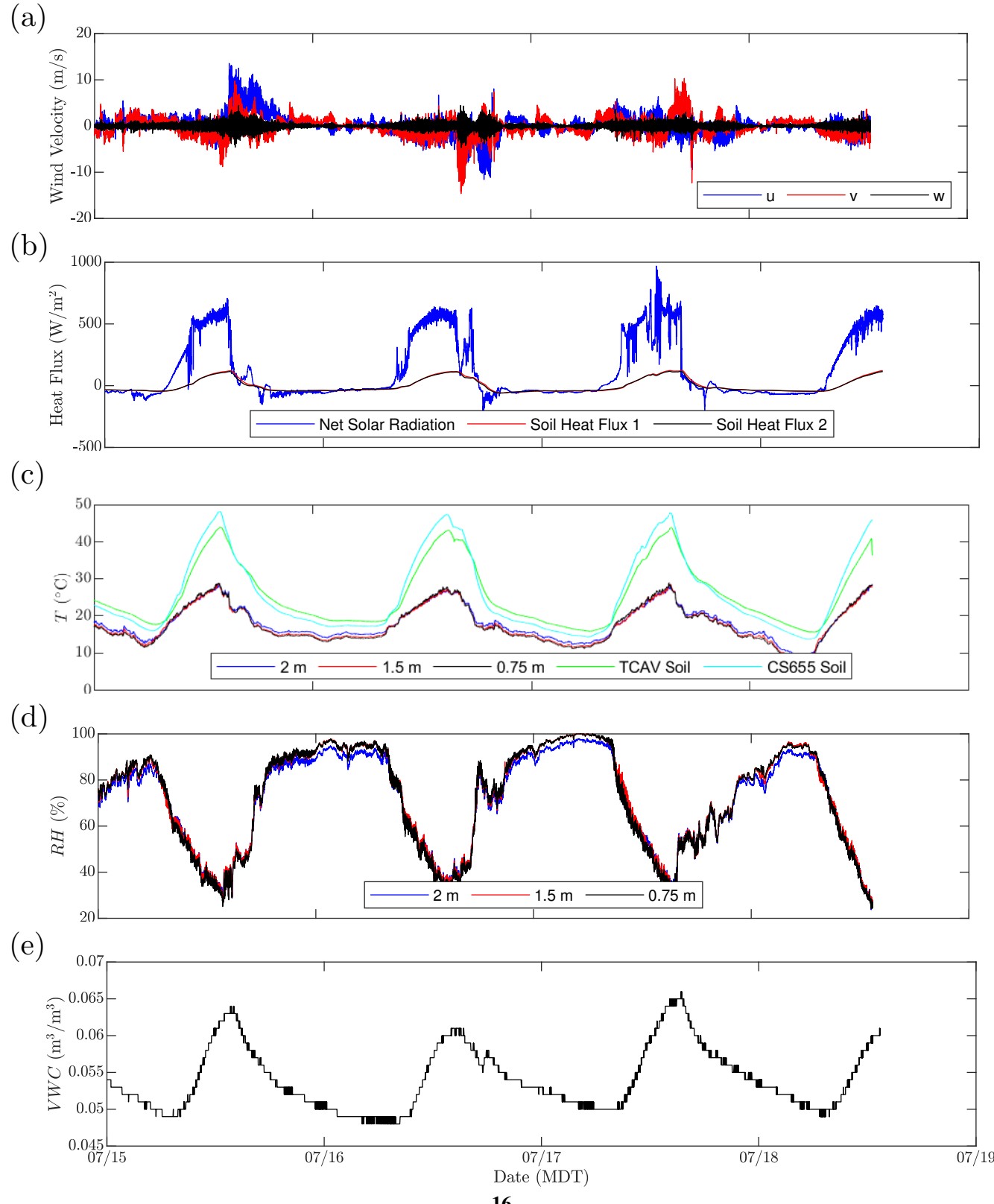

**Figure 6.** Flux tower measurements of: (a) wind velocity components in East $u$, North $v$, and vertical $w$ directions; (b) net solar radiation and soil heat flux; (c) temperature, $T$; (d) relative humidity, $RH$; and (e) soil volumetric water content, $VWC$.

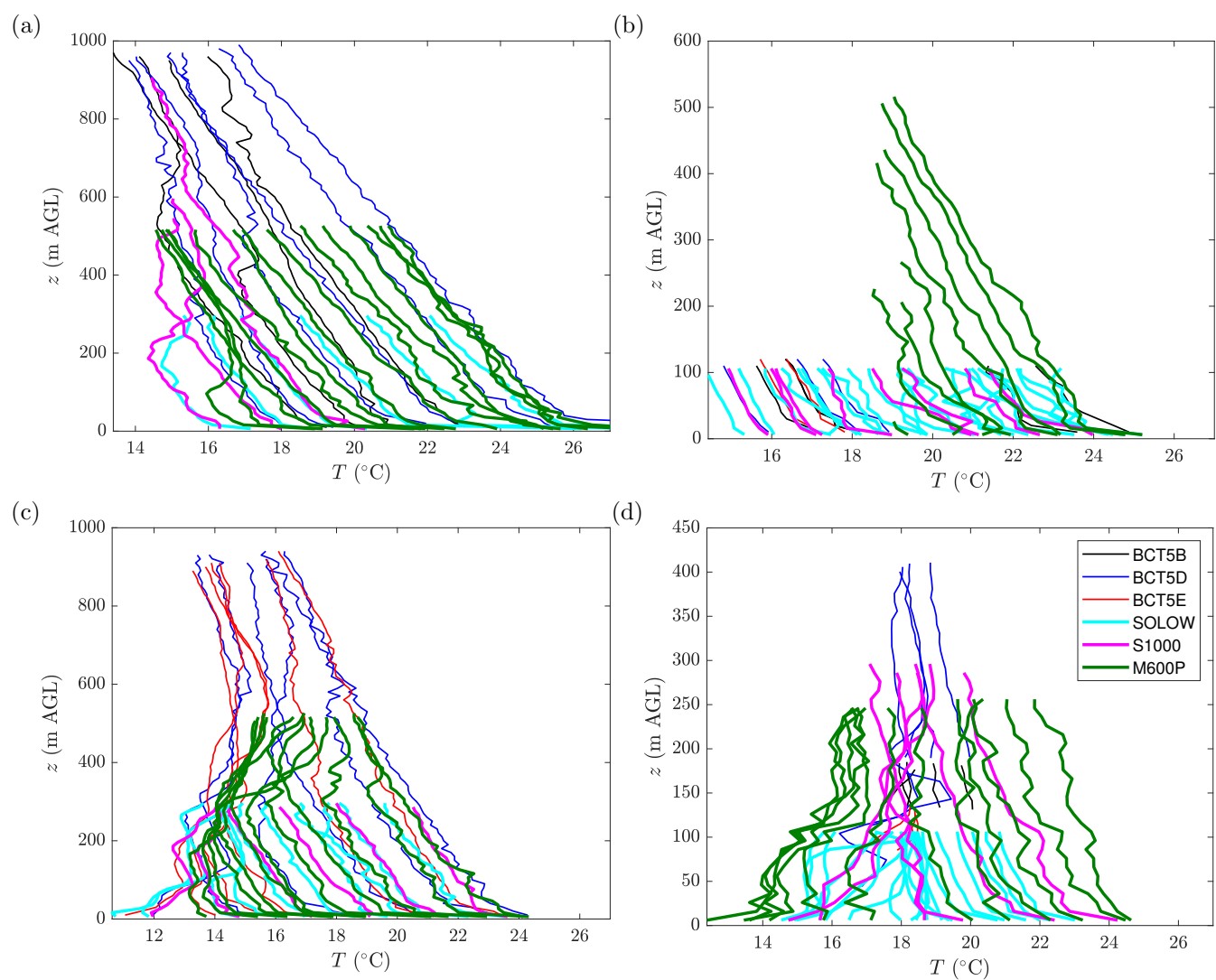

**Figure 7.** Temperature measured by aircraft on: (a) July 15, 2018; (b) July 16, 2018; (c) July 18, 2018; and (d) July 19, 2018. Note different axis heights for each day. Profiles have been produced by averaging over 10 m height intervals from each flight.

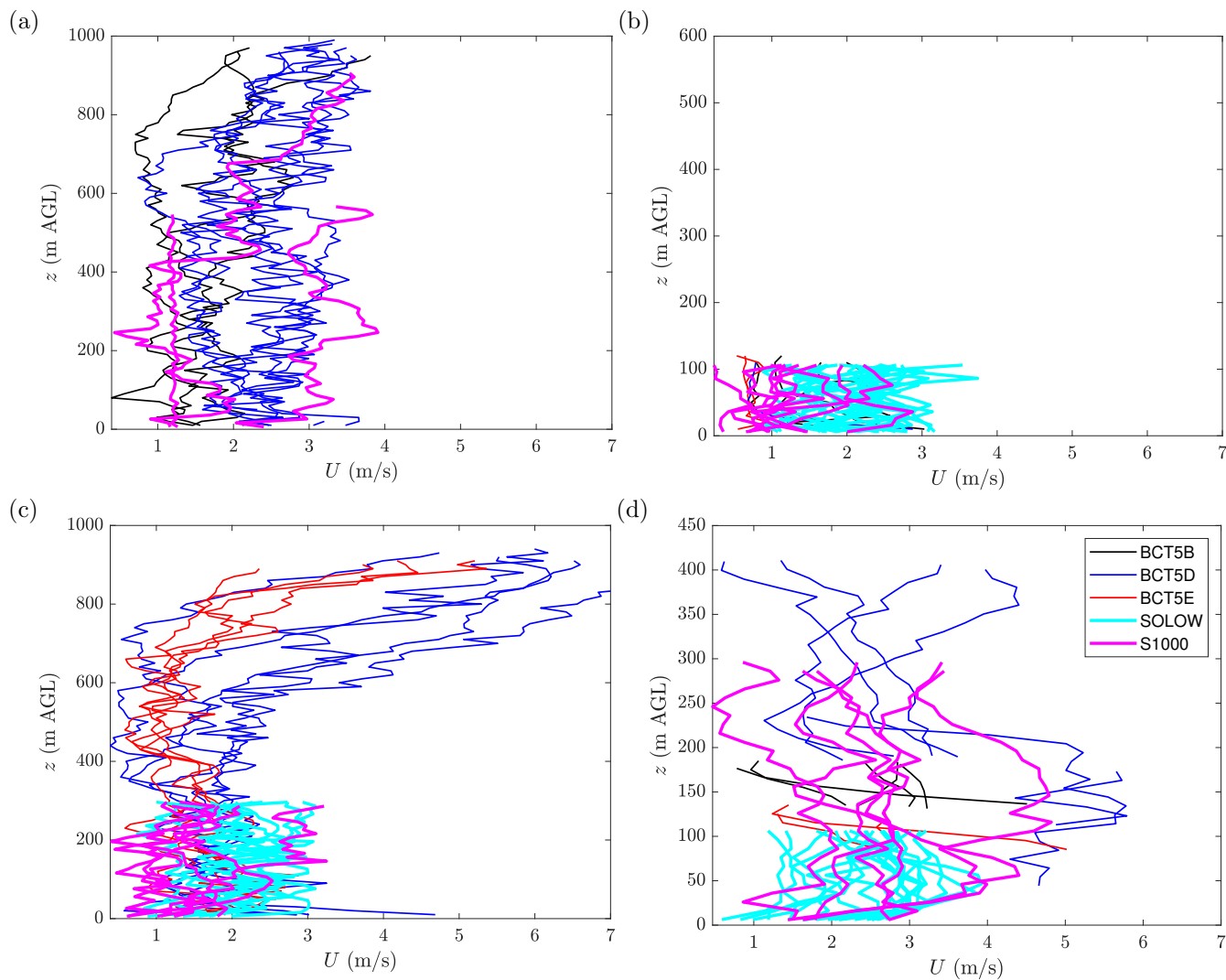

**Figure 8.** Wind velocity magnitude: (a) July 15, 2018; (b) July 16, 2018; (c) July 18, 2018; and (d) July 19, 2018. Axis heights same as Figure 7. Profiles have been produced by averaging over 10 m height intervals from each flight.

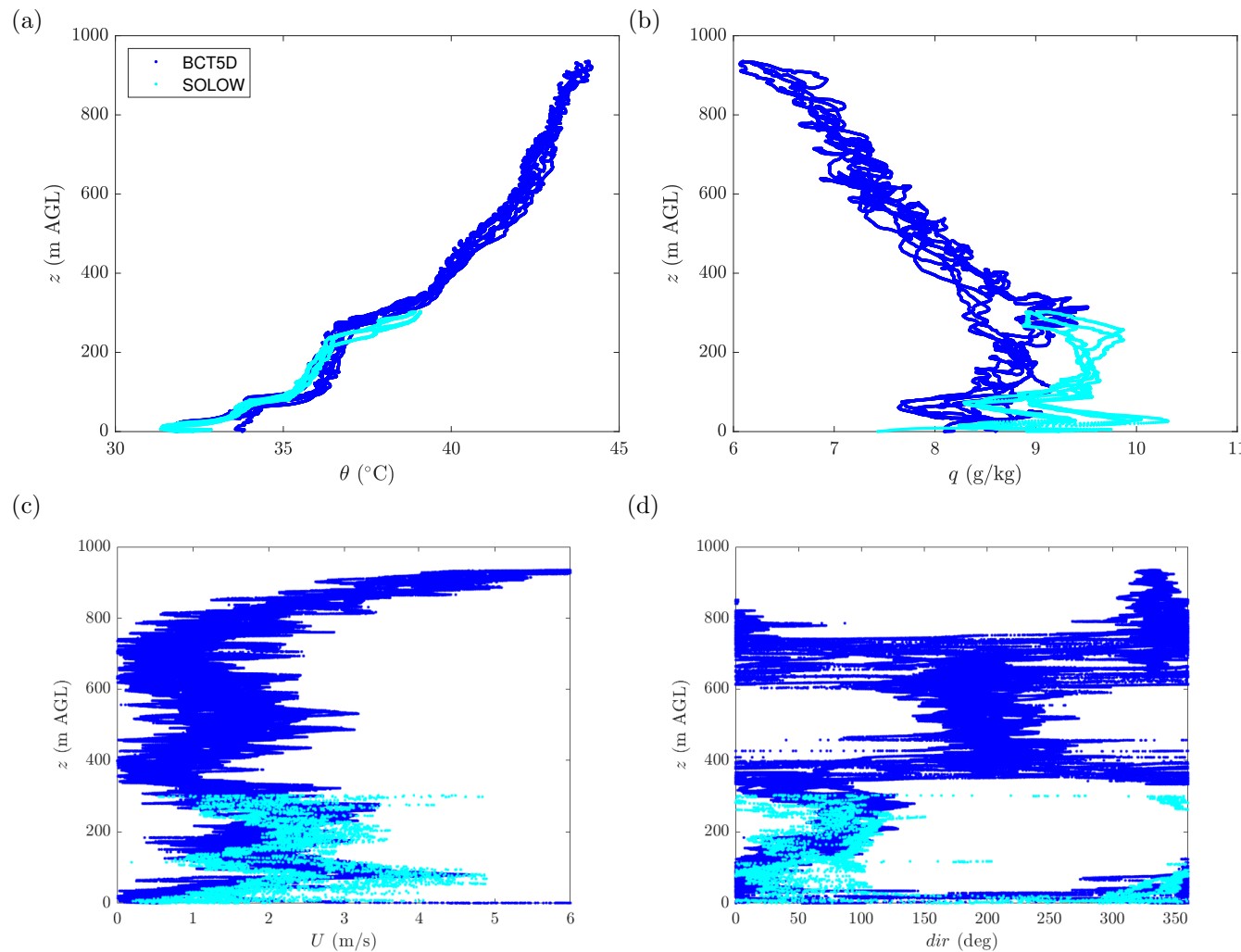

**Figure 9.** Example vertical profiles measured between 7:00 MDT and 7:45 MDT by BCT5D and SOLOW on July 18, 2018 at location Kilo. Potential temperature profiles shown in (a), water vapor mass mixing ratio profiles shown in (b), with wind speed, $U$, and direction, $dir$, profiles shown in (c) and (d) respectively.

statistics, as illustrated in the sample profiles from some of the earliest co-located flights on that day which are presented in Figure 9. In Figure 9, profiles from the SOLOW rotorcraft are shown for a 10 minute flight initiating at 7:00 MDT and profiles from BCT5D are shown for a flight initiating at 7:15 MDT that lasted 30 minutes. Hence the time period represented by these profiles is approximately 45 minutes and any features evident persisted over that duration. Note that, despite this time difference, no dependence of measured values on ascent or descent can be observed in these measurements and there appears to be good agreement between the different aircraft and sensor systems.

The potential temperature profiles, shown in Figure 9(a), suggests that the boundary layer was largely stable, except for a thin region of unstable air near the surface. Significant changes in the potential temperature, $\theta$ gradient are evident at 50 m AGL, 100 m AGL, 250 m AGL and 300 m AGL. The corresponding water vapor mass mixing ratio, $q$, profiles, shown in Figure 9(b), are slightly less complex, but changes in the vertical gradient are also observed at 50 m AGL, 100 m AGL and 250 m AGL coinciding with the changes observed in the potential temperature. The wind horizontal magnitude and direction are shown in Figure 9(c) and (d) respectively. These figures show that the winds also experienced significant vertical and horizontal shearing. For the lowest 100 m AGL, winds were from the north at up to 3 m/s, whereas they were from the east at 2 m/s from 100 m AGL to 300 m AGL. From 300 m AGL to 600 m AGL they were from the south at 2 m/s and above that altitude the winds were from the north, increasing in magnitude with altitude until reaching 6 m/s at 900 m. Notably, these trends were evident on both aircraft systems at altitudes where overlapping measurements are available. Similar trends were observed for other UAS throughout the morning during the same period with mixed layer conditions establishing up to 300 m AGL at 9:15 MDT, up to 500 m AGL at 10:15 MDT, 600 m AGL at 11:15 MDT and throughout the measurement domain at 12:00 MDT.

## 7 Conclusions

In July 2018 in the San Luis Valley in Colorado USA, researchers from multiple institutions participated in the Lower Atmospheric Profiling Studies at Elevation - a Remotely-piloted Aircraft Team Experiment (LAPSE-RATE) measurement campaign. As part of this campaign effort, University of Kentucky researchers contributed seven fixed-wing and rotorcraft unmanned aerial systems, totaling 178 successful data acquisition flights over four days. For three of the four days, the flights consisted of profiles measured up to altitudes of 900 m above ground level. The goal of these flights was to observe the boundary layer state during morning transition within the elevated valley, and to identify precursors for convection initiation. For the fourth day the gravity driven flow into the valley was measured using three fixed-wing and one rotorcraft aircraft conducted transects at six altitudes while the remaining aircraft conducted vertical profiling.

The data from these coordinated UAS flights provide significant contribution to the characterization of the lower atmosphere within the valley during the LAPSE-RATE campaign. All data from the systems described here are now distributed and can be freely accessed from the Zenodo data repository (https://doi.org/10.5281/zenodo.3701845). Although some preliminary quality control and bias correction have been conducted on this data, some caveats for their use have been described in this manuscript. These caveats include noting that data from the BC5T and SOLOW aircraft are sampled at rates exceeding that of the response for their respective sensing systems. In addition, wind data from the rotorcraft should only be considered valid on the ascent.

## 8 Data availability

The data files for each flight from each aircraft are available from the Zenodo open data repository (https://doi.org/10.5281/zenodo.3701845) (Bailey et al., 2020). This data consists of 178 files containing thermodynamic and kinematic data (pressure, temperature, humidity, wind speed and direction) measured by the unmanned aerial systems and fixed 2 m flux tower operated by the University

of Kentucky during the LAPSE-RATE campaign. These data have undergone preliminary quality control in the form of bias corrections and elimination of some false readings. Files are posted for each individual UAS flight in netCDF format. For the flux tower data, each measurement day is presented as a separate netCDF file.

The netCDF files have each variable listed individually with self-describing metadata to provide information on the source and units for the data. Missing data points or those determined to have bad values have been set to -9999.9.

Files are named using the following standard: UKY.ppppp.b1.yyyymmdd.hhmmss.cdf with ppppp being the 5-letter platform identifier, yyyymmdd is the file date (UTC year, month, day, month) and hhmmss the file start time (UTC hours, minutes, seconds).

*Author contributions.* The University of Kentucky contribution to LAPSE-RATE was planned and coordinated by SWS, MPS, JBH, MIG and SCCB, flight team members included CAC, LFP, ZSL, TJS, JDH, SBM, ISR, CDS, VGS, CNV and HMW, data analysis and preparation was conducted by SCCB, with senior team members SCCB, SWS and MPS contributing to preparation of this manuscript.

*Competing interests.* The authors declare no competing interests are present.

*Acknowledgements.* This work was supported by the US National Science Foundation through award no. CBET-1351411 and by award
no. 1539070, Collaboration Leading Operational UAS Development for Meteorology and Atmospheric Physics (CLOUDMAP). Additional student travel support was provided through NSF AGS 1807199 and DOE DE-SC0018985.

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
