# Peer review of "University of Kentucky measurements of wind, temperature, pressure and humidity in support of LAPSE-RATE using multi-site fixed-wing and rotorcraft UAS"

_Earth System Science Data, 2020_

## Referee Comment (RC1) · Anonymous Referee #1 · 14 May 2020

General comments

The manuscript by Bailey et al. introduces the data set available from UAS deployed during the LAPSE-RATE experiment by the University of Kentucky. Four different platforms, either fixed-wing or rotorcraft measured boundary layer parameters as temperature, relative humidity, pressure, and wind. The manuscript provides access to the data set and clearly defines the accuracy of each sensor, the different flights, and the associated technical and regulatory limitations. Quality control and bias correction are also implemented. Description of the UAS are well-documented, yet this reviewer

recommends a table to improve the access to the information. However, a comprehensive overview of the dataset is missing. A section should be added to provide figures showing an overview of the variables. In addition, the time series of the meteorological measurements would allow an assessment of the weather status during which the UAS measurements were taken. Statistic figures (profiles, histograms, etc.) of each variable would help to identify interesting meteorological periods for further scientific analysis. The figures also need to be improved before final publication (for example, maps and flight plan displays).

Specific comments

Page 2 Section 2: It would be complementary to the text to add pictures of each platform with their sensors to visualize the placement of instruments on the UAS.

A table including the UAS and the sensor description, accuracy and resolution should be added. The flux tower instrumentation should also be included in the table.

Page 6 Figure 1: In the legend, add the meaning of the colored dots. It is not clear where each UAS flew on each day when just looking at Figure 1. Incorporating information from table 2 would improve the figure.

Page 7 Table 2: This table should also describe the flight pattern associated to each UAS mission for each day.

Page 8 line 177: For the fixed-wing, what is the diameter of the spirals? For fixed-wing and rotorcraft, what are the ascent/descent rates?

Page 9 Figure 2: Add the legend for blue and red profiles in the figure. Does Zulu represent the 'Zulu time'? Is it a location? Would is rather be location Kilo? A map should be incorporated in Figure 2(a) to show the terrain associated with the location even if the profiles are also similar for other locations on July 16 and 18. Add also the transects from BCT5B in a similar figure as Figure 2(a) with a map. The transects are not easily identified in Figure 1(b).

Page 10 line 213: Do horizontal profiles correspond to transects? What is the length of the transects?

Page 11 Figure 3: The topography in Figure 3(a) is helpful, however the flight patterns are not visible. An appropriate scale should be selected to emphasize the different flights. In Figure 3(b), for the legend with the names of the UAS, same color and same order should be kept between Figures (a) and (b). Different markers or linewidth would clarify which aircraft is a fixed-wing or a rotorcraft UAS.

Page 13 Figure 4: What is the location Zulu? Would it rather be location Kilo? Add the legend for red and blue lines in the figure.

Page 12 Section 5: A case study is provided in this section; however, as the broader meteorological context is not introduced, it is not straight forward how to interpret these UAS measurements and identify relevant periods for further analysis. An overview of the data set is needed, such as time series of the meteorological conditions monitored by the tower over the four days. Statistic figures (temperature, relative humidity, mixing ratio, wind, etc.) should be provided to summarize the flights on each day at each location.

Page 14 line 266: "USA researchers from multiple institutions", credit also needs to be given to researchers from other countries and foreign institutions.

Technical corrections

Page 2 line 25: Remove one 'of'

---

## Referee Comment (RC2) · Anonymous Referee #2 · 31 May 2020

The manuscript provides an overview of data collected by the University of Kentucky during the LAPSE RATE campaign. Data were collected by several types of UAS including one type of fixed-wing vehicle and three types of multi-rotor vehicles. In addition, data were collected on a small meteorological tower at the surface.

The manuscript is short and well-written and the description of the various platform and instruments is quite detailed. I consider this manuscript a data report. This is the intention of Earth System Science Data papers if I understand correctly from the description of the scope of the journal. There is no significant scientific content in this

paper. I have some specific comments listed below.

1) Providing pictures of the various UAS used with details of the location of the various sensors would be very useful for the data users

2) I liked Figure 3b and these figures should also be included for the other flight days. In addition, tables with timing of each flight should be provided in my opinion for a data report.

3) Figure 3a is not very clear. I think that zooming in the area of the flight operations would make the flightpaths much clearer. Perhaps a 2D, rather than 3D map would also make things clearer.

4) Ascent and descent rates of the UAS (in particular the multi-rotor UAS) AND their justification should be provided. Also, were these rates kept constant every time for the multi-rotor UAS?

5) Line 96: how was the optimal mast height determined? Perhaps it was not so 'optimal' given the bias that was found (in section 4, see also later comment)

6) Line 227-230: This needs some clarification: a.Expand on 'acceptance range violations' b.What 'intermittency' of occurrences? I thought that the multirotor UAS ascended and descended in a continuous fashion? c.If data were not removed were they at least flagged in the data files?

7) Can the temperature and humidity data collected during a descent be used without any correction (due to e.g. slow time response of sensors)? Please provide an example figure in which temperature and humidity are plotted for ascent and descent during one flight

8) Line 240-244: These biases represent a major weakness in this data set. As a scientist potentially interested in using these data, I am not satisfied by the action taken by the authors that "These biases have been removed in the data files made openly available". Much more detailed information need to be provided about possible

causes. Was there no dependency of the bias on the wind speed? Was the same bias present during hovering and ascending flights? How about wind direction? Was the bias equal in u and v component of the wind? Anyhow, at this point, without any additional information, I would be very skeptical about using the data.

9) line 245: Mention somewhere what time sunrise was on July 18.

10) Minor typo: remove duplicate 'of' in line 25.

---

## Author Comment (AC1) · 1 Jul 2020

Response to Referee Report #1 for
**University of Kentucky measurements of wind, temperature, pressure and humidity in support of LAPSE-RATE using multi-site fixed-wing and rotorcraft UAS**
submitted to Earth System Science Data

Thank you for taking the time to review and report on our manuscript. We have made

appropriate revisions to our original manuscript submission (which are indicated in the manuscript using blue text) and provide point-by-point responses to individual comments below. For convenience the original Referee comments are provided in italics with our responses added below:

*The manuscript by Bailey et al. introduces the data set available from UAS deployed during the LAPSE-RATE experiment by the University of Kentucky. Four different platforms, either fixed-wing or rotorcraft measured boundary layer parameters as temperature, relative humidity, pressure, and wind. The manuscript provides access to the data set and clearly defines the accuracy of each sensor, the different flights, and the associated technical and regulatory limitations. Quality control and bias correction are also implemented. Description of the UAS are well-documented, yet this reviewer recommends a table to improve the access to the information. However, a comprehensive overview of the dataset is missing. A section should be added to provide figures showing an overview of the variables. In addition, the time series of the meteorological measurements would allow an assessment of the weather status during which the UAS measurements were taken. Statistic figures (profiles, histograms, etc.) of each variable would help to identify interesting meteorological periods for further scientific analysis. The figures also need to be improved before final publication (for example, maps and flight plan displays).*

Figure 1, in concert with Tables 1 and 2 were intended to provide the information required to access specific information, providing Day/time/location and aircraft information. Please keep in mind that the dataset being described in this paper comprises 178 individual files, each providing measurements of 3 or more thermodynamic/kinematic variables. This means providing information about the contents of each individual file in a tabulated format is not efficient and therefore providing a comprehensive overview of the dataset is a challenging enterprise. In addition, this manuscript is submitted as part of a special edition detailing the LAPSE-RATE campaign, and an overview of the meteorological conditions and measurement objectives will be provided by de Boer et

al., with the intent of helping to identify interesting meteorological periods for further scientific analysis.

That said, we have added an additional table providing additional information about the resolution and uncertainty of the variables measured by all systems. We have also added an additional section presenting time series of the variables measured by the flux tower, as well as additional figures presenting all measured profiles of temperature and wind magnitude. We feel that this information should provide a sufficient overview of the measurement results, and are currently conducting more focused statistical analysis of the data as part of more detailed investigation of the results. Note that the figures in the compiled .pdf did not reflect the original quality of the .png images. We have replaced these images with .eps versions which we hope will compile better in the final production.

Specific comments:

1. *Page 2 Section 2: It would be complementary to the text to add pictures of each platform with their sensors to visualize the placement of instruments on the UAS.*

   We have added photographs of each of the aircraft and the flux tower as Figures 1 and 2 respectively.

2. *A table including the UAS and the sensor description, accuracy and resolution should be added. The flux tower instrumentation should also be included in the table.*

   This table has been added as Table 3.

3. *Page 6 Figure 1: In the legend, add the meaning of the colored dots. It is not clear where each UAS flew on each day when just looking at Figure 1. Incorporating information from table 2 would improve the figure.*

   We apologize for not including this information in the legend and originally intended this figure to be used in conjunction with Table 2. Please note that the

colored dots are the actual flight trajectories for each day overlaid on the topographical map. We have updated the legend and tried to improve the overall clarity of the figure.

4. *Page 7 Table 2: This table should also describe the flight pattern associated to each UAS mission for each day.*

   We have added this information to the table.

5. *Page 8 line 177: For the fixed-wing, what is the diameter of the spirals? For fixed-wing and rotorcraft, what are the ascent/descent rates?*

   This information has been added to the text.

6. *Page 9 Figure 2: Add the legend for blue and red profiles in the figure. Does Zulu represent the 'Zulu time'? Is it a location? Would is rather be location Kilo? A map should be incorporated in Figure 2(a) to show the terrain associated with the location even if the profiles are also similar for other locations on July 16 and 18. Add also the transects from BCT5B in a similar figure as Figure 2(a) with a map. The transects are not easily identified in Figure 1(b).*

   You are correct, this should have said Kilo not Zulu. Also, we have changed Figure 2 to follow a recommendation of Referee 2 and it now presents the cadence for all days and all flights, providing a more comprehensive summary of the entire dataset. For conciseness, the three-dimensional flight profiles are no longer shown. Note that the terrain is flat for all locations except Poison Gulch and Saguache Airfield (see Figure 1) and therefore providing a map to correspond with these flights does not provide much additional information beyond what is already provided in Figure 1.

7. *Page 10 line 213: Do horizontal profiles correspond to transects? What is the length of the transects?*
Yes, we internally use the term horizontal profiles interchangeably with transects. The transects were nominally 1700 m long on July 19th with two 13000 m long transects conducted on July 16th. For clarity we have revised the manuscript to replace the term 'horizontal profiles' with 'transects' measurements and added the typical transect length to the text.

8. *Page 11 Figure 3: The topography in Figure 3(a) is helpful, however the flight patterns are not visible. An appropriate scale should be selected to emphasize the different flights. In Figure 3(b), for the legend with the names of the UAS, same color and same order should be kept between Figures (a) and (b). Different markers or linewidth would clarify which aircraft is a fixed-wing or a rotorcraft UAS.*

We have updated Figure 3(a), replacing it with two figures to better illustrate the flight profiles at the Poison Gulch and Saguache Airfield measurement sites. Note that the same colors were already used between the two figures with the exception of the M600 which was changed to improve visibility due to the different backgrounds in figure 3a vs 3b. We have updated all figures in the manuscript to ensure consistent coloring between figures. Although not as clear as we would have liked, we have also made the rotorcraft lines thicker than those of the fixed-wings. We found that increasing/decreasing line thickness further detracted from the readability of the figures.

9. *Page 13 Figure 4: What is the location Zulu? Would it rather be location Kilo? Add the legend for red and blue lines in the figure.*

You are correct. We have fixed this in the manuscript.

10. *Page 12 Section 5: A case study is provided in this section; however, as the broader meteorological context is not introduced, it is not straight forward how to interpret these UAS measurements and identify relevant periods for further analysis. An overview of the data set is needed, such as time series of the mete-orological conditions monitored by the tower over the four days. Statistic figures*

*(temperature, relative humidity, mixing ratio, wind, etc.) should be provided to summarize the flights on each day at each location.*

As noted above, we have added an additional section presenting time series of the variables measured by the flux tower, as well as additional figures presenting all measured profiles of temperature and wind magnitude. Although we understand why the referee would like such a presentation, we feel that additional statistical analysis of the results is beyond the intended scope of the current manuscript. Note that to present a separate figure for each location and day would require 13 separate figures per variable (not including the flux tower results) and that a full comprehensive presentation as suggested by the referee would significantly increase the manuscript length.

11. *Page 14 line 266: "USA researchers from multiple institutions", credit also needs to be given to researchers from other countries and foreign institutions.*

   Note that the full text "... in the San Luis Valley in Colorado, USA researchers from multiple institutions..." was written with the intent for the the "USA" to refer to the location of the measurement, not the origin of the researchers involved. We have revised the text by moving the comma after the USA to be more clear in our intended meaning.

Technical Corrections:

1. *Page 2 line 25: Remove one 'of'*
   Removed.

---

## Author Comment (AC2) · 1 Jul 2020

Response to Referee Report #2 for
**University of Kentucky measurements of wind, temperature, pressure and humidity in support of LAPSE-RATE using multi-site fixed-wing and rotorcraft UAS**
submitted to Earth System Science Data

Thank you for taking the time to review and report on our manuscript. We have made

appropriate revisions to our original manuscript submission (which are indicated in the manuscript using blue text) and provide point-by-point responses to individual comments below. For convenience the original referee comments are provided in italics with our responses added below each one:

Specific comments:

1. *Providing pictures of the various UAS used with details of the location of the various sensors would be very useful for the data users*

   We have added photographs of each of the aircraft and the flux tower as Figures 1 and 2 respectively.

2. *I liked Figure 3b and these figures should also be included for the other flight days. In addition, tables with timing of each flight should be provided in my opinion for a data report.*

   We have replaced Figure 2 with a new figure showing the flight cadence for each day.

   Please keep in mind that the dataset being described in this paper comprises 178 files corresponding each flight. This makes providing information about individual flights in a tabulated format inefficient and lengthy.

3. *Figure 3a is not very clear. I think that zooming in the area of the flight operations would make the flightpaths much clearer. Perhaps a 2D, rather than 3D map would also make things clearer.*

   We have split figure 3a into two separate figures to improve clarity and better illustrate flight trajectories as we can zoom in closer to the aircraft locations. Due to overlapping flight trajectories, we feel the 3D plot is a better illustration of the spatial coverage provided by these flights.
4. *Ascent and descent rates of the UAS (in particular the multi-rotor UAS) AND their justification should be provided. Also, were these rates kept constant every time for the multi-rotor UAS?*

   This information has been added to the text (2-3 m/s for the SOLOW and 1 m/s for the S1000). The justification has also been added which was to provide a balance between minimizing vertical resolution and maximizing the number of individual profiles for the SOLOW and maximizing the vertical resolution of a single profile while measuring to a higher altitude for the S1000.

5. *Line 96: how was the optimal mast height determined? Perhaps it was not so 'optimal' given the bias that was found (in section 4, see also later comment)*

   We have revised the text to note that the optimal height was found by increasing the post length until running the the UAS motors at full speed did not result in a change in reading. Note that bias described in Section 4 is due to the sonic anemometer design itself and is dependent on the the flow angle relative to the anemometer. For the intercomparison measurements described in Barbieri et al. (2019), in which the aircraft was hovering at the height of a reference system on a mast (rather than profiling), the measured velocities tracked very well with that of the reference system.

6. *Line 227-230: This needs some clarification: a.Expand on 'acceptance range violations' b.What 'intermittency' of occurrences? I thought that the multirotor UAS ascended and descended in a continuous fashion? c.If data were not removed were they at least flagged in the data files?*

   a. The acceptance range violations are the instances when the cone angle exceeded the $\pm 30°$ limitation described in section 2.2. The text has been modified to clarify this point. b. As the horizontal wind magnitude was variable, there would be flights when the net direction vector would be within the acceptance cone when the induced vertical velocity due ascent/descent was imposed on the

wind vector. Note that, the descent velocity was slower (2 m/s) which resulted in the improved reliability of the results from this flight direction. c. We had not initially flagged the data, but due to the reviewer's comment, we have uploaded a new set of data with the ascents flagged with -9999.9.

7. *Can the temperature and humidity data collected during a descent be used without any correction (due to e.g. slow time response of sensors)? Please provide an example figure in which temperature and humidity are plotted for ascent and descent during one flight*

Note that this example figure was already provided as Figure 4 for both fixed-wing and rotorcraft and showed no dependence of measured values on ascent or descent for either temperature or humidity for either system. In addition, this figure showed good agreement between the different aircraft. Note that in the revised manuscript, these profiles are Figure 9.

8. *Line 240-244: These biases represent a major weakness in this data set. As a scientist potentially interested in using these data, I am not satisfied by the action taken by the authors that "These biases have been removed in the data files made openly available". Much more detailed information need to be provided about possible causes. Was there no dependency of the bias on the wind speed? Was the same bias present during hovering and ascending flights? How about wind direction? Was the bias equal in u and v component of the wind? Anyhow, at this point, without any additional information, I would be very skeptical about using the data.*

We agree that these biases are of some concern, which is why they are explicitly mentioned in the text, such that researchers interested in using this data do so with full understanding of their presence. We have added additional context in the text. Note that their removal does result in satisfactory intercomparison between different aircraft (as presented in Figure 9, for example) as well as satisfactory intercomparison with the independent systems of the University of Colorado MURC (presented in the intercomparison study of Barbieri et al.), and ASOS system at Saguache airport (manuscript under preparation). As we intentionally remove them uniformly without adjustment in any way on either a day-to-day or per-flight basis, and their removal results in agreement between multiple types of instruments and platforms, we feel their source is intrinsic sensor bias such that their removal is justified and does not detract from the usefulness of this dataset.

9. *line 245: Mention somewhere what time sunrise was on July 18.*

   Sunrise and sunset times have been added to the description of the area of operations.

10. *Minor typo: remove duplicate 'of' in line 25.*

    Removed.